# Pib2 as an Emerging Master Regulator of Yeast TORC1

**DOI:** 10.3390/biom11101489

**Published:** 2021-10-09

**Authors:** Riko Hatakeyama

**Affiliations:** Institute of Medical Sciences, School of Medicine, Medical Sciences & Nutrition, University of Aberdeen, Foresterhill, Aberdeen AB25 2ZD, UK; riko.hatakeyama@abdn.ac.uk

**Keywords:** Pib2, TOR, mTOR, TORC1, mTORC1, yeast

## Abstract

Cell growth is dynamically regulated in response to external cues such as nutrient availability, growth factor signals, and stresses. Central to this adaptation process is the Target of Rapamycin Complex 1 (TORC1), an evolutionarily conserved kinase complex that fine-tunes an enormous number of cellular events. How upstream signals are sensed and transmitted to TORC1 has been intensively studied in major model organisms including the budding yeast *Saccharomyces* *cerevisiae*. This field recently saw a breakthrough: the identification of yeast phosphatidylInositol(3)-phosphate binding protein 2 (Pib2) protein as a critical regulator of TORC1. Although the study of Pib2 is still in its early days, multiple groups have provided important mechanistic insights on how Pib2 relays nutrient signals to TORC1. There remain, on the other hand, significant gaps in our knowledge and mysteries that warrant further investigations. This is the first dedicated review on Pib2 that summarizes major findings and outstanding questions around this emerging key player in cell growth regulation.

## 1. Introduction: Major Players in TORC1 Regulation

Whether uni- or multicellular, all organisms flexibly adapt to their ever-changing environment at the individual cell level. The signaling cascades allowing dynamic and strict adjustment of cell growth are of fundamental importance. In eukaryotes, at the center of such signaling pathways is the evolutionarily conserved Target of Rapamycin Complex 1 (TORC1), the molecular target of the macrolide compound rapamycin as its name suggests [1,2,3]. TORC1, of which the catalytic subunit is the TOR protein kinase, is activated and inhibited, respectively, by pro-growth signals such as nutrients and growth factors and by anti-growth stresses. TORC1, in turn, activates pro-growth cellular processes such as anabolic reactions (e.g., nucleotide/lipid/protein synthesis) while repressing catabolic reactions (e.g., protein degradation via autophagy). Not surprisingly given the prevalence and importance of processes regulated by this kinase, dysregulation of mammalian/mechanistic TORC1 (mTORC1) is associated with many human diseases including cancer, diabetes, and neurodegeneration as well as aging [4]. TOR also forms structurally and functionally distinct complex, TORC2 [5,6], which I do not cover in this article.

One of the most intensively studied areas in the TORC1 field is its upstream regulatory mechanisms. This line of research is important not only in understanding how cells sense and interpret growth-affecting signals but also in developing means to manipulate mTORC1 activity, which potentially enables us to treat mTORC1-related diseases or even extend our life span. In identifying TORC1 regulators, various model systems have been deployed. Among them, the budding yeast *Saccharomyces cerevisiae* (hereafter simply ‘yeast’) occupies a unique position; the powerful genetic toolsets for this relatively simple eukaryote allowed the initial identification of Rag small GTPases as major regulators of TORC1 [7], prior to its confirmation in higher eukaryotes [8,9], similarly to the initial identifications of TOR itself [10] and the other components of TORC1 [11] in the past. A recent example of such yeast-led breakthroughs is the identification of Pib2, a novel critical regulator of TORC1 [12].

Before diving deeper into Pib2, the main theme of this review, I outline other key regulators of TORC1 (Figure 1), because their relation to Pib2 is an important area of future research. The above-mentioned Rag small GTPases, Gtr1 and Gtr2 in yeast (corresponding to RagA/B and RagC/D in mammals, respectively), are established major regulators of TORC1 [13,14]. The heterodimer of Rag GTPases localizes to the endolysosomal membranes by forming an extended complex, called the EGO complex in yeast and the Rag-Ragulator complex in mammals, together with conserved scaffolding proteins. The nucleotide-binding status of Rag GTPases depends on the nutrient (e.g., amino acids) availability; in nutrient-rich conditions, Gtr1/RagA/B and Gtr2/RagC/D are loaded with GTP and GDP, respectively, forming the TORC1-activating conformation. The immediate upstream regulators of Rag GTPases, e.g., GTPase-activating proteins (yeast SEACIT complex [15,16] and mammalian GATOR1 complex [17] for Gtr1/RagA/B; yeast Lst4-Lst7 complex [18] and mammalian FNIP-Folliculin complex [19] for Gtr2/RagC/D) are generally well conserved [20]. On the other hand, the uppermost signaling components, i.e., amino acid sensors, appear to have diverged between yeast and mammals [21].

In mammals, the Rag-Ragulator complex (in its mTORC1-activating form) recruits mTORC1 to the lysosomal surface, allowing its access to another small GTPase Rheb [22]. Rheb then stimulates the catalytic activity of TORC1 in an allosteric manner [23,24,25]. Rheb itself gets GTP-loaded and thereby activated by growth factors such as insulin [26]. The sequential action of Rag and Rheb can be understood as a mechanism to integrate nutrient and growth factor signals, allowing cells to make a unified decision on their growth. How the yeast EGO complex regulates TORC1 is less clear; the Rag-Rheb sequential model is unlikely to be true in yeast, as the yeast Rheb does not seem to activate TORC1.

TORC1 regulation by Rag GTPases is bidirectional; upon nutrient starvation, the Rag heterodimer comprised of GDP-loaded Gtr1/RagA/B and GTP-loaded Gtr2/RagC/D inhibits TORC1. This inhibitory regulation is also common to yeast and mammals, though the mechanisms appear distinct [27,28]. Interestingly, the bidirectional nature of TORC1 regulation is shared by Pib2 (see Section 5.4 for details), potentially reflecting the need to fine-tune the TORC1 activity in a delicate manner.

Apart from Rag and Rheb GTPases, many other molecules have been reported to act upstream of TORC1, of which examples in yeast include, but are not limited to, the AMP-activated kinase Snf1, the translation initiation factor kinase Gcn2, and the Whi2-Psr1/Psr2 phosphatase complex [29,30,31,32]. However, it is not trivial to define direct regulators of TORC1 because TORC1 responds to virtually any cellular stress, and lacking a random protein can easily affect TORC1 activity, indirectly, by causing certain stress. Another critical, but often overlooked factor, is the nutrient uptake rate; nutrients are transported into cells by specific transporter proteins operating at the plasma membrane, which undergo multi-layered regulation by various transcription factors, post-translational modifications and the intracellular trafficking machinery [33]. Altered activity of any nutrient transporter, by whatever means, can significantly affect the TORC1 activity. This might be the case for Whi2-Psr1/Psr2, for example, as this complex regulates the uptake of, at least, ammonium [34]. After excluding proposed TORC1 regulators that have not been extensively studied at a mechanistic level, and therefore remain possible to indirectly affect TORC1 activity either by protecting cells from stresses or by modulating the nutrient uptake, arguably, Rag GTPases have long been the only well-established, dedicated signaling component that directly and critically regulates TORC1 in yeast.

The ‘Rag-only era’ has continued from 2005, when the EGO complex was discovered [7], till 2015, when Pib2 started being characterized [12]. This work by Kim and Cunningham, together with the following independent studies by multiple groups including ours, has established Pib2 as a direct master regulator of TORC1 [32,35,36,37,38,39,40]. It would not be an exaggeration to describe Pib2 as one of the two most prominent regulators of yeast TORC1 identified so far. Despite its evident importance in the TORC1 pathway and the significant amount of information already collected, there is currently no dedicated review on Pib2. I, therefore, summarize in this article major findings and outstanding questions around Pib2, to further stimulate future studies on this emerging key player in cell growth regulation.

## 2. Pib2 before 2015

Pib2, of which the systematic gene name is YGL023C, stands for the PhosphatidylInositol(3)-phosphate (PI3P) Binding protein 2. It was named in 2001 [41] because of the presence of a so-called FYVE zinc finger domain, a well-characterized PI3P-binding motif [42,43]. PI3P is a membrane lipid particularly enriched in endosomes and vacuoles (yeast counterparts of lysosomes) [44], where TORC1 resides [45,46]. However, the mere identification of the FYVE domain in Pib2 did not immediately spark its characterization in the context of TORC1 signaling (in fact, in any context).

Years later, in 2005, a hint on the Pib2 function popped up in the seminal genetic screening that led to the identification of the EGO complex [7]. Dubouloz et al. in De Virgilio’s group conducted a genome-wide screening to search for gene-deletion mutants that cannot recover from the growth arrest induced by rapamycin, a TORC1 inhibitor. Such mutants were expected to lack a gene that is necessary to reactivate TORC1. There, along with the mutants lacking the EGO complex components, the Pib2-lacking mutant (*pib2∆*) appeared within top hits, suggesting a critical role of Pib2 in TORC1 activation. However, their following studies mainly focused on the EGO complex.

Another unbiased survey published in 2011 also implied a Pib2-TORC1 link. A phospho-proteomic analysis identified Pib2 as a potential substrate of Npr1 kinase, a prominent downstream effector of TORC1 [47]. Pib2 thus seemed to act both upstream and downstream of TORC1. Although these pieces of information implied a close relation to TORC1, focused and extensive investigations on Pib2 were not initiated for some time.

## 3. Identification of Pib2 as a TORC1 Regulator

2015 marked the birth year of Pib2 research in the context of TORC1 signaling. Kim and Cunningham started working on Pib2 [12] because it shows sequence similarity to human LAPF/phafin1 protein, which promotes lysosomal membrane permeabilization [48], the phenomenon they have been studying. After confirming that Pib2 promotes the same phenomenon in yeast, they naturally proceeded to characterize its molecular function. Because Pib2 was an almost uncharacterized protein, they took unbiased bioinformatic approaches; they performed hierarchical clustering analysis on previously published genome-wide datasets of chemical and genetic interaction profiles. They discovered that Pib2 clusters with known TORC1 pathway components such as the subunits of TORC1 itself or the EGO complex, suggesting a tight link between Pib2 and TORC1. Indeed, they observed a slightly decreased TORC1 activity in *pib2∆* cells, confirming Pib2’s positive role in TORC1 signaling.

An important, debate-stimulating fact highlighted in the Kim and Cunningham’s paper was the synthetic lethality between *pib2∆* and the EGO complex deletants. This is clearly caused by the fatally low TORC1 activity, because the synthetic lethality was rescued by the expression of a hyperactive TOR allele [12]. These observations initially prompted many to draw a simple model, in which Pib2 and the EGO complex independently activate TORC1 (as described in Figure 1). This parallel model was later supported by the observed complete loss of TORC1 activity upon simultaneous depletion of Pib2 and the EGO complex, as well as biochemical data showing that Pib2 and the EGO complex form complexes with distinct pools of TORC1 in a mutually exclusive manner [38].

However, there also exist observations that do not easily fit the parallel model. First, Pib2 shows physical interaction with the EGO complex in multiple yeast two hybrid-based systems [12,49]. Second, lacking either Pib2 or the EGO complex alone causes the entire loss of the acute TORC1 activation by amino acids [35,39], rather than a partial defect that the parallel model would predict. These data favor an alternative model in which Pib2 and the EGO complex work cooperatively [39].

This apparent controversy in the Pib2-EGO relationship remains a major mystery in the field. I propose a model that can reconcile many observations in Section 4.3, but do not exclude other possibilities. It should be noted that each study chose different experimental settings (e.g., yeast strain background, TORC1 activating stimulus, and TORC1 activity readout), making the direct comparison difficult. For example, different auxotrophies of laboratory strains potentially diversify the results, because Gcn2 kinase inhibits TORC1 only when an auxotrophic strain is starved for amino acids it is auxotrophic for [30,32]. It can also be problematic to compare the results obtained with different readouts, because TORC1 may differentially regulate distinct downstream processes depending on characteristics, the localization for example [46], of substrate proteins. The methods and results of the relevant studies are summarized in Table 1.

## 4. Pib2 in Amino Acid Response

Having rigidly established Pib2 as a major regulator of TORC1 [12], Kim and Cunningham’s work left a question on the nature of physiological signals that Pib2 transmits. Shortly after, studies from two independent groups including ours unraveled the critical role of Pib2 in the TORC1 activation by glutamine [35,36]. The same was shown for leucine and ammonium [32,39]. Since then, Pib2 has mainly been studied in the context of amino acid/nitrogen signaling.

### 4.1. Pib2 as a Glutamine Sensor

The study by Tanigawa and Maeda [36] was particularly informative mechanism-wise because it took in vitro approaches. They developed two unique in vitro TORC1 kinase assays that use permeabilized, cytosol-free cells or purified vacuoles as the TORC1 source. In their system, glutamine and cysteine, but not other amino acids, activated TORC1 in a Pib2-dependent manner. This amino acid selectivity was in line with their in vivo data suggesting that Pib2 is more important for TORC1 activation by glutamine than for that by arginine [36]. An important conclusion from their work was that the vacuoles alone contain all the essential machinery for the Pib2-mediated TORC1 activation by glutamine and cysteine.

The situation was different for the EGO complex. In their in vitro system, the EGO complex failed to contribute to TORC1 activation [36], implying that essential component(s) of the amino acid-EGO-TORC1 branch is missing in permeabilized cells and purified vacuoles.

Very recently, the same group proceeded to demonstrate that even bacterially purified Pib2 interacts with and activates purified TORC1 in response to glutamine (although activation was detectable only when Pib2 carries a hyperactivating mutation [37]). Pib2 alone is therefore sufficient to transmit the glutamine signal to TORC1; in other words, Pib2 turned out to be a long-sought cellular glutamine sensor acting upstream of TORC1. Its novelty and importance are not necessarily compromised by preceding identification of multiple amino acid sensors in mammalian cells [50,51], as the situation is significantly different in yeast, a prototrophic organism, unlike mammalian cells that cannot self-synthesize all the required amino acids [21].

### 4.2. Dual-Phase Activation of TORC1

When discussing the amino acid-induced TORC1 activation mechanism, it is worth taking into account the dual-mode cellular response discovered by Stracka et al. in Hall’s group [52] (Figure 2). By carefully following the time-course of TORC1 activity after amino acid stimulation, they observed at least two distinct phases of TORC1 activation; the first phase of activation (hereafter Phase-1) was acute (starting within 1 min) and transient (ceasing within 5 min or so), while the second one (Phase-2) was slow (starting in 15 min) and continuous.

Importantly, the two phases showed different amino acid selectivity [52]; Phase-1 was induced by virtually any amino acids, while Phase-2 was observed for only a subset of them, particularly the ones known as good-quality nitrogen sources such as glutamine [33]. It appears as if, in Phase-2, TORC1 is monitoring the richness of nitrogen sources (likely by measuring as an indicator some key metabolite in core nitrogen metabolism, possibly glutamine [52]), resulting from metabolic circuits starting from a given single amino acid. It makes sense that, for the long-term cell growth, what matters is the total richness of nitrogen sources rather than the repertoire of individual amino acids, because wild-type yeast is capable of converting any amino acid to another via metabolic reactions.

In Phase-1, on the other hand, TORC1 may be monitoring each amino acid species before they are metabolized. In support of this possibility, even unmetabolizable, thus non-growth-promoting amino acid analogues do activate TORC1 transiently [53]. Saliba et al. in Andre’s group proposed an interesting model that the exact readout cells utilize there is the proton influx catalysed by the plasma membrane-localized amino acid/proton symporters. The proton pump Pma1 seems to mediate this signal [53,54], but how it is transmitted to TORC1 is currently unclear. It should also be noted that this model was challenged by others; Brito et al. reported that a pH drop is not a prerequisite for TORC1 activation [32].

A physiological advantage of having two phases can be speculated in the following way. When starved cells are exposed to amino acids, they first make an instant decision to switch on (or prepare) the pro-growth cellular programs, without waiting for the consequences of metabolic reactions (Phase-1). At this point, cells do not know if they are going to eventually obtain enough nutrients to sustain their growth; they just predict that it may be the case. However, making a quick, prediction-based decision should be advantageous as it allows them to start growth earlier than the surrounding competitor cells/organisms. Then, once they are assured that they have accumulated enough nutrients after metabolic reactions, they undergo the actual continuous growth (Phase-2). The TORC1 activities in Phase-1 and -2 can be regarded as the gearshift and the accelerator pedal of a car, respectively.

### 4.3. Pib2 and the EGO Complex in the Two Phases

An important finding in the study by Stracka et al. was that the EGO complex is indispensable only for Phase-1 [52]; yeast lacking the EGO complex, while showing the complete lack of the acute and transient activation, eventually activated TORC1, almost normally, 15 min after the amino acid stimulation. This could explain why the EGO deletants show only a minor defect in the basal TORC1 activity in a bulk cell culture, where the contribution of Phase-2 to the total TORC1 activity should dominate over that of Phase-1 (which occurs only transiently upon nutrient uptake). Also, this is perhaps why the EGO deletants show an almost normal rate of cell growth once they start growing; their problem lies rather in initially waking up from the growth arrest [7]. All these observations are common to *pib2∆* [35,39] (also our unpublished results), potentially explaining why this mutant showed only a minor defect in the TORC1 activity in the initial Kim and Cunningham’s study [12]. Therefore, while both Pib2 and the EGO complex are strictly required for Phase-1, the two appear to work redundantly in Phase-2. Consistently, the steady-state TORC1 activity completely diminishes upon conditional, simultaneous depletion of Pib2 and the EGO complex [38].

The different Pib2/EGO dependencies suggest that TORC1 is activated via distinct mechanisms in Phase-1 and -2. Theoretically, the observations can be best explained by the following model; Pib2 and the EGO complex work together in Phase-1, while independently in Phase-2. This two-mode model, while speculative, can reconcile many observations which are otherwise puzzling or conflicting. As discussed in Section 3, there are observations that either support or oppose the parallel relationship between Pib2 and the EGO complex. The supporting ones, such as the *pib2∆*-EGO*∆* synthetic lethality [12], the formation of distinct Pib2-TORC1 and EGO-TORC1 supercomplexes [38] and the Pib2-EGO redundancy in the basal TORC1 activity [38] can all be reflecting the events during the steady-state cell growth, which should largely reflect the continuous, Phase-2 TORC1 activity. In contrast, opposing evidence, the complete lack of acute TORC1 activation in single deletants [35,39] clearly concerns Phase-1. The physical interaction between Pib2 and the EGO complex was observed in yeast two-hybrid systems but not in co-immunoprecipitation assays [12]; this could be because the two-hybrid system, where the readouts are foot-prints of protein-protein interactions, may be more sensitive in capturing transient interactions such as those occurring only in Phase-1.

The in vitro TORC1 activation monitored in Maeda’s studies [36,37] shares characteristics with the in vivo activation in Phase-2, rather than that in Phase-1 [52]; it showed (1) slow kinetics (Phase-1 is acute and very strong), (2) no requirement of the EGO complex (which is strictly required in Phase-1), and (3) the amino acid selectivity to glutamine and cysteine (Phase-1 is not that selective). Their assays might be capturing the Pib2-TORC1 signaling branch in Phase-2, and therefore, yet another approach might be needed to reconstitute the Phase-1 activation in vitro.

The facts and speculations around Phase-1 and -2 are summarised in Figure 2. As already mentioned, there can be alternative views and interpretations. For example, the observed lack of Phase-1 in certain strains or conditions can rather be interpreted as a delayed response. Also, the generality of this dual-phase model is currently unclear, as it is largely based on the phosphorylation kinetics of a single TORC1 substrate, Sch9 kinase [52].

## 5. Domain Structure of Pib2

The Pib2 protein can be divided into 4 distinct functional modules (Figure 3) [12,35]. To discuss the mode of Pib2’s action in more details, here I summarize the known facts for each domain.

### 5.1. FYVE Domain (426–532 Amino Acid Residues)

As mentioned in Section 2, the presence of a FYVE PI3P-binding motif had been known [42], based on the amino acid sequence, before people started specifically working on Pib2. As expected, the FYVE domain is required for the Pib2 localization to PI3P-rich organelles, i.e., endosomes and vacuoles [12].

The Pib2 mutant lacking the FYVE domain is partially functional [35]. Therefore, while the membrane recruitment of Pib2 might help its efficient interaction with TORC1 residing on the same membrane, it is not absolutely required for activating TORC1.

The FYVE-PI3P binding might not be the only determinant of Pib2 localization. The endosomal localization of Pib2 is limited to the specific subpopulation coined signaling endosomes (SE), the distribution pattern shared by TORC1 and the EGO complex [46,55]. Although not much is known on the lipid composition of SE membrane, SE, as well as vacuoles, are particularly enriched with the membrane lipid phosphatidylinositol 3,5-bisphosphate (PI(3,5)P_2_) in addition to PI3P [56]. PI(3,5)P_2_ is synthesized from PI3P by the Fab1 lipid kinase [57]. Because Pib2 interacts with TORC1, and TORC1 interacts with PI(3,5)P_2_ via its Kog1 subunit [58], it is possible that the Pib2-TORC1 supercomplex preferentially binds to the membrane enriched with both PI3P and PI(3,5)P_2_, which interacts with Pib2 via its FYVE domain and with TORC1 via its Kog1 subunit, respectively (Figure 3). The phenomenon that multiple factors determine the membrane recruitment of proteins is quite common and known as the ‘coincidence detection’ [59].

### 5.2. Kog1-Binding Domain (KBD) (165–425 Residues)

N-terminally next to the FYVE domain lies the largely disordered, Kog1-binding domain that mediates the Pib2-TORC1 interaction. This interaction involves at least two binding surfaces, one residing within 165–304 residues and the other within 312–430 residues [35]. Studies from Noda’s and Maeda’s groups revealed a critical role of the 337–341 residues in this interaction, because mutations in this region significantly weakened the Pib2-TORC1 interaction as well as TORC1 activity [37,38]. The interaction between Pib2 and TORC1 is strengthened by glutamine both in vivo and in vitro [36,37,38], likely representing a critical step in the glutamine-induced TORC1 activation.

The 304–533 residues, which include the FYVE domain and a part of the KBD domain, is sufficient for the glutamine-responsive TORC1 binding [37]. This region must therefore contain the glutamine recognition site(s), which has not been further defined.

### 5.3. C-Terminal TORC1-Activating Domain (CAD) (533–635 Residues)

The C-terminal end, particularly the last 15 amino acids, is essential for TORC1 activation by Pib2 [12,35]. Reflecting its functional importance, this region is well conserved among fungal species and even in human LAPF/phafin1 [12].

Tanigawa et al. in Maeda’s group isolated hyperactive Pib2 alleles harboring mutations in the 626–629 residues, which enabled them to detect direct activation of TORC1 by Pib2 in vitro [37]. This may imply the existence of an intra-domain autoinhibition mechanism, though its physiological significance is unclear.

Uncovering how the CAD domain works is obviously key in understanding the mechanism of TORC1 activation. This domain alone does not interact with Kog1 at least in the yeast two-hybrid system [35], thus it may work on another TORC1 subunit, the catalytic subunit Tor1 for example. Interestingly, this domain of Pib2 was needed for TORC1 to form an agglomerate (‘TORC1-body’) upon nutrient starvation [40]. The proper interpretation of this observation perhaps awaits the definitive characterization of the TORC1-body and its relation to the TORC1 activity, which are currently under an active debate [29,40,60]. In any case, solving the structure of the Pib2-TORC1 supercomplex will greatly help us define the TORC1 activation mechanism, as the Rag-mTORC1 and the Rheb-mTORC1 complex structures did [25,61].

### 5.4. N-Terminal Inhibitory Domain (NID) (1–164 Residues)

An unexpected discovery was made in an innovative transposon-based screening developed by Michel et al. in Kornmann’s group [35]. There, more and less frequent transposon insertion to a specific genomic locus indicates a disadvantageous and advantageous effect, respectively, of that particular locus on cell fitness in a given culture condition. In accordance with its positive role on the TORC1 activity, the *C*-terminal regions of Pib2 saw less frequent transposon insertion in the presence of rapamycin. In contrast, strikingly, more frequent insertion was observed for its N-terminal regions. This result suggested a TORC1-inhibiting role for the Pib2 N-terminus, which we confirmed by truncation analysis [35]. The first 50 amino acids are sufficient for this inhibitory effect [38]. How NID inhibits TORC1 is unclear, but it works independently from CAD and the EGO complex [35]. It is unknown in which physiological context this domain functions, as it seems not to be involved in the amino acid response [35].

## 6. Regulation of Pib2

As summarised so far, a significant amount of information has been reported on how Pib2 regulates TORC1. Contrary, much less is known on how Pib2 is regulated, a question of fundamental importance in understanding how cells sense growth-affecting signals upstream of TORC1.

An immediate question is how glutamine stimulates the Pib2-TORC1 interaction. The direct interaction with glutamine increases the thermal stability of Pib2 [37]. However, further structural analyses are needed to uncover the exact nature of the Pib2-glutamine interaction, and how it facilitates the interaction with TORC1. Also, there may exist additional factors involved in this process; Ukai et al. in Noda’s group reported that the Pib2-glutamine interaction is enhanced by preincubating Pib2 with yeast cell lysate [38].

Amino acids are not the only nutrient source that TORC1 reacts to. Pib2 also contributes to TORC1 activation by ammonium, another nitrogen source [32]. This could be explained by the Pib2’s function as a glutamine sensor [37], as ammonium seems to activate TORC1 after being assimilated into glutamine [52]. Carbon sources, such as glucose, is another critical determinant of TORC1 activity [29]. Pib2 is involved in the regulation of TORC1-body formation (see Section 5.3) by glucose [40], but mechanistically how Pib2 receives glucose signal is unknown.

Whether and how Pib2 conveys any stress signals to TORC1 remains an open question. Because its NID is not required for the amino acid response [35], it is possible that Pib2 transmits other signal(s)/stress(es) using this domain.

As mentioned in Section 2, Npr1, a TORC1 downstream kinase, promotes Pib2 phosphorylation [47]. This suggests the existence of a feedback regulatory loop (TORC1-Npr1-Pib2-TORC1), which was confirmed by a following study [32]. What this phosphorylation does on Pib2, mechanistically, is unclear. Because Npr1 inhibits TORC1 in a Pib2-dependent manner [32], one possibility (not experimentally proven) is that phosphorylation of Pib2 activates its NID-mediated TORC1 inhibitory function. The existence of multiple feedback regulations has been known for the TORC1 pathway [62], not surprisingly given the role of this pathway as a homeostatic controller of cellular metabolism.

The relationship between Pib2 and the EGO complex may vary depending on upstream signals/stresses; they might transmit different, but partially overlapping repertoires of signals, which are eventually integrated by TORC1. It is important to investigate how Pib2 reacts to signals other than amino acids, and how it orchestrates with other TORC1 regulators including the EGO complex.

## 7. Pib2 in Other Organisms

To date, analysis of Pib2 in the context of TORC1 signaling has been limited to those in the budding yeast model. The most closely related, structurally, human protein is LAPF/phafin1, that shares at least the FYVE and CAD domains with yeast Pib2 [12]. They also share a positive role in lysosomal membrane permeabilization, which is mediated by TORC1 at least in the Pib2’s case [12,48]. Moreover, LAPF/phafin1 regulates autophagy and endolysosomal morphology [63], processes controlled by TORC1 [56,64,65]. Despite these structural and functional similarities between Pib2 and LAPF/phafin1, direct evidence is currently missing for the mTORC1 regulation by LAPF/phafin1 [66].

Pib2 is well conserved in fungi [12]. If the Pib2-TORC1 signaling is not conserved in humans, Pib2 would become a promising antifungal drug target. It is therefore worth evaluating Pib2’s contribution to the pathogenicity of infectious fungi such as *Candida* species. Studying the apparent Pib2 homolog of the fission yeast *Schizosaccharomyces pombe* [12] will offer a unique opportunity; one can address its relation to Rheb, of which the TORC1-activating function is conserved in this organism [67], unlike in budding yeast.

Some of the downstream functions of TORC1, for example in protein metabolism, are well conserved in plants [68]. Moreover, plant TORC1 responds to amino acid signals as yeast and mammalian TORC1 do [69]. However, plants seem to have very different TORC1 regulatory mechanisms [70] due to the surprising lack of both Rag and Rheb GTPases [71]. It would be interesting to know whether Pib2 turns out to be the first direct regulator of TORC1 conserved across fungi, animals and plants.

## 8. Final Remarks: What’s Next?

Research into Pib2 began only 6 years ago. Nevertheless, since then, significant contributions from multiple groups have solidly established Pib2 as one of the most prominent regulators of yeast TORC1. Yet, many fundamental questions and controversies, such as the ones scattered in this review, are waiting to be solved. Those include: (1) How does glutamine interact with Pib2, and promote its interaction with TORC1? (2) How do, mechanistically, the CAD and NID domains work? (3) What is the relationship with the EGO complex (parallel or cooperative)? (4) Does Pib2 transmit any signal other than amino acids/nitrogen sources? (5) How is Pib2 feedback-regulated by the TORC1-Npr1 cascade? (6) To which extent is the Pib2-TORC1 signaling evolutionarily conserved? Continuing efforts of scientists in the coming decade will surely provide answers to many of these important questions, further expanding the horizon of Pib2 research, and thereby advancing our understanding of cell growth regulation.

## Figures and Tables

**Figure 1 biomolecules-11-01489-f001:**
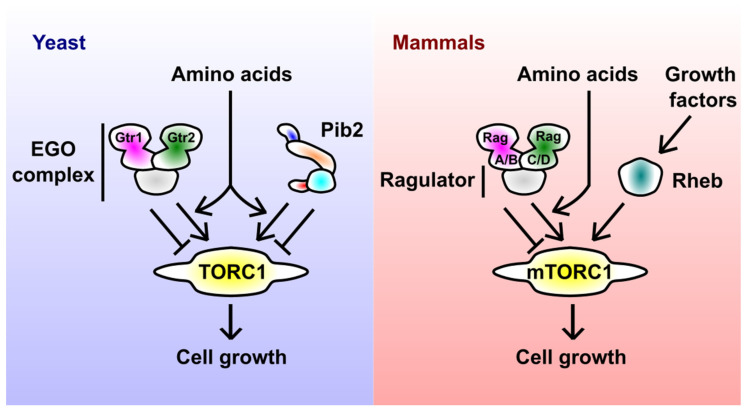
Major regulators of yeast and mammalian TORC1. In yeast, amino acids activate TORC1 via the EGO complex (consisting of the Gtr1-Gtr2 small GTPase heterodimer and scaffolding subunits) and Pib2. The schematic representation of Pib2 reflects its domain structure (see Section 5 for details). In mammals, amino acid and growth factor signals to mTORC1 are mediated by the Rag-Ragulator complex (the EGO complex equivalent) and Rheb small GTPase, respectively.

**Figure 2 biomolecules-11-01489-f002:**
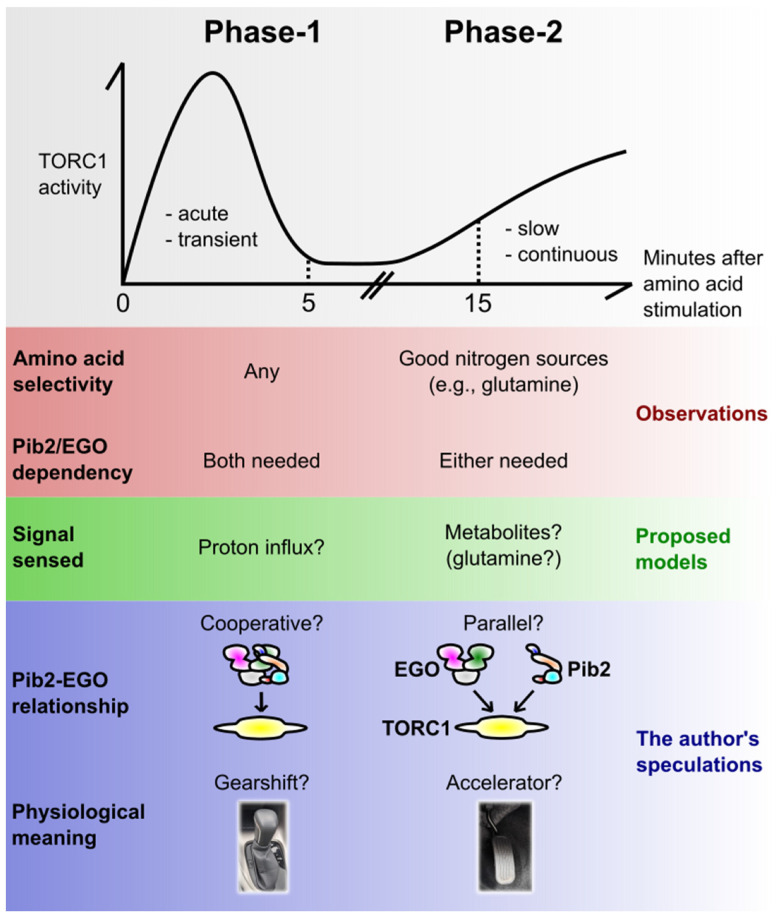
The dual-phase activation of yeast TORC1 by amino acids. Yeast TORC1 responds to amino acids with two distinct phases over time: the acute, transient pulse (here coined Phase-1) and the slow, continuous activation (Phase-2). This figure summarizes the observed distinct natures, proposed or speculative mechanistic models for the two phases. See the main text for details.

**Figure 3 biomolecules-11-01489-f003:**
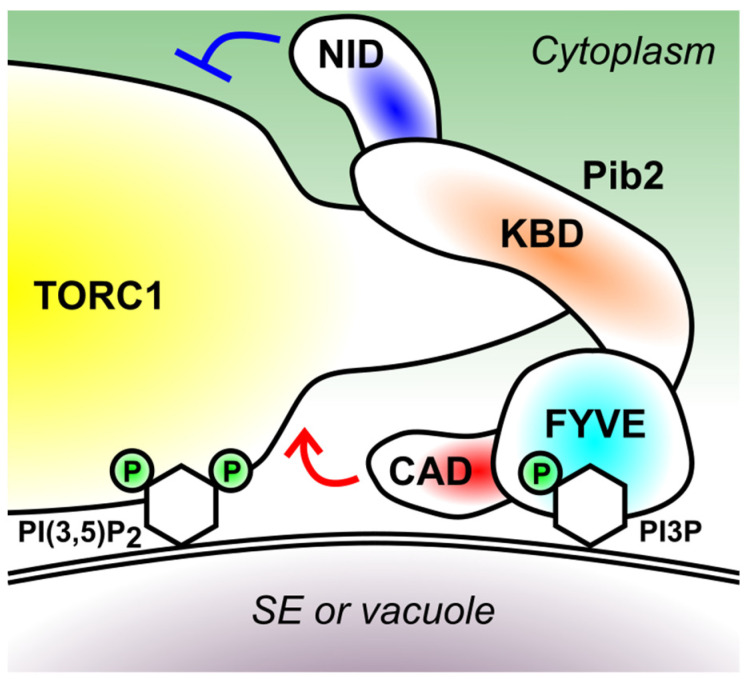
The domain structure of Pib2 and a model of its membrane recruitment. Pib2 is recruited to the surface of signaling endosomes (SE) and vacuoles via the interaction between its FYVE domain and the membrane lipid PI3P. The KBD domain mediates the interaction with the Kog1 subunit of TORC1, which in turn interacts with the membrane lipid PI(3,5)P_2_. ‘P’ denotes phosphorylation on the inositol rings (depicted as hexagons) of PI3P and PI(3,5)P_2_. The CAD and NID domain of Pib2 activates and inhibits TORC1, respectively.

**Table 1 biomolecules-11-01489-t001:** Observations supporting either the Pib2-EGO parallel or cooperative model.

Observation	Supported Model	Yeast Strain Background	Culture Media/ TORC1 Stimuli	TORC1 Readout	References
*pib2∆*-EGO*∆* synthetic lethality	Parallel	BY4741/4742 (auxotroph)	YPD	Viability (rescued by hyperactive TORC1)	[12]
Pib2-EGO redundancy in the basal TORC1 activity	Parallel	BY4741 (auxotroph)	SD with casamino acid	Sch9 phosphorylation (C-terminus band-shift)	[38]
Partial defects in TORC1 activation (>10 min) in *pib2∆* and EGO*∆*	Parallel	Σ1278b (prototroph) [32]; S288C (prototroph) [36]	Minimal proline medium +glutamine or ammonium stimulation [32]; SD-N +glutamine stimulation [36]	Rps6 phosphorylation (phospho-specific antibody) or Npr1 phosphorylation (band-shift) [32]; Sch9 phosphorylation (phospho-specific antibody) [36]	[32,36]
Distinct Pib2-TORC1 and EGO-TORC1 supercomplexes (co-immunoprecipitation)	Parallel	BY4741 (auxotroph)	YPD	N/A	[38]
Pib2-EGO interaction (yeast two-hybrid)	Cooperative	N/A	N/A	N/A	[12,49]
Strict requirement of both Pib2 and EGO for acute and transient TORC1 activation (<5 min)	Cooperative	BY4741/4742 (prototroph) [35]; S288C (prototroph) [36]; W303A (auxotroph) [39]	Minimal proline medium +glutamine stimulation [35]; SD-N +glutamine or arginine stimulation [36]; SD-N +glutamine or leucine stimulation [39]	Sch9 phosphorylation (phospho-specific antibody) [35,36]; Rps6 phosphorylation (phospho-specific antibody [39]	[35,36,39]

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
