# Peer review of "Pib2 as an Emerging Master Regulator of Yeast TORC1"

_biomolecules, 2021, doi:10.3390/biom11101489_

Round 1

Reviewer 1 Report

This review article entitled “Pib2 as an emerging master regulator of yeast TORC1” by Riko Hatakeyama is about Pib2 dependent regulatory mechanism of TORC1. In fact this could be the first review on Pib2. The necessary topics are mostly comprehensively covered and the author’s original insight is provided. One critical concern is that unpublished paper from Maeda lab was cited as a pivotal topic. As it seems to be already accepted, publication of this review should be awaited until Maeda paper is publicly open.

Minor points;

  1. Reference #2 #6 #8 #34 #41 should be amended.

Author Response

I thank the reviewer for taking time reading the manuscript and providing encouraging and helpful comments.

Specific points:

This review article entitled “Pib2 as an emerging master regulator of yeast TORC1” by Riko Hatakeyama is about Pib2 dependent regulatory mechanism of TORC1. In fact this could be the first review on Pib2. The necessary topics are mostly comprehensively covered and the author’s original insight is provided. One critical concern is that unpublished paper from Maeda lab was cited as a pivotal topic. As it seems to be already accepted, publication of this review should be awaited until Maeda paper is publicly open.

Response: The paper by Tanigawa et al (Reference 37) is now available.

Minor points;

  1. Reference #2 #6 #8 #34 #41 should be amended.

Response: Corrected. Thank you for spotting this.

Reviewer 2 Report

In this timely review, Hatakeyama provides comprehensive perspective on the relationship between TORC1 and Pib2, a proposed glutamine sensor for TORC1 in yeast.  The review is straightforward to read and provides a nice historical perspective to contextualize this burgeoning field, and I think this is a timely contribution.  I have only small comments throughout that are intended to improve on the manuscript (no major revisions recommended).

I recommend providing the full name for Pib2 in the abstract (PI3P-Binding Protein 2) for those of us less familiar with the yeast literature.

Throughout, studies are described using the senior author’s name, which is certainly helpful for understanding the historical context of experiments; as a matter of principle, I think it is worthwhile to also mention the first author’s name.  So, for example, l. 113: “De Virgilio’s group” should be “Dubouloz et al. in De Virgilio’s group”; l. 163: “Maeda’s group” should be “Tanagawa and Maeda”, etc.

Introduction section: Given the emphasis on evolutionary comparison, I think it is worth mentioning here that Pib2 is conserved in many fungi but not well-conserved outside fungi.  I also think it is worth noting that, in plants, TOR is sensitive to amino acids (see, e.g., Schaufelberger et al., J Experimental Botany, 2019; Cao et al., eLife, 2019; O’Leary et al., Plant Cell, 2020; and Liu et al., Dev Cell, 2021) and has conserved roles in translation and ribosome biogenesis downstream from TOR (see, e.g., Xiong et al., Nature, 2013, and Scarpin et al., eLife, 2020), but that plants do not encode Gtr1/Gtr2 orthologues (see, e.g., Chantranupong et al., Cell, 2015; and Brunkard, Dev Cell, 2020).  I think this would help to strengthen the argument that there is diversity in TOR nutrient-sensing mechanisms across eukaryotes, and that the singular emphasis on Rag/Gtr GTPases is perhaps too narrow (even in humans).

The review often refers readers to other sections (e.g., ll. 137-138; but this happens at least ten times).  While I understand the impulse, this can make reading the review choppy: a reader should be able to read from top to bottom.  If it is important to foreshadow an in-depth discussion below, then this is OK, but in general, I don’t think it is necessary to always refer to reader to future paragraphs.  I would delete ll. 137-138, for example.

References are occasionally missing; see, for example, l. 176, which I think should refer to citation #33.

Repeatedly, the review refers to a citation and then also to unpublished results from the author’s group.  I recommend against this, because it potentially undercuts the published work.  Instead, this could be phrased as, “which our lab has replicated/reproduced”, if the goal is to indicate that you strongly believe a conclusion is valid.  In general, citing unpublished work or preprints in reviews seems risky to me.

l.176-178: I’m not sure that Pib2 is sufficient to transmit the glutamine signal. The work in citation #33 (Ukai et al., 2018, PLoS Genetics) argues that Pib2 is necessary for glutamine sensitivity and that Pib2-TORC1 interact with glutamine, but even they note that other proteins may be involved and that Pib2 may not be the sensor per se. We know from other examples in mammals that necessary components of amino acid sensor complexes are not necessarily the amino acid-interacting proteins.  I would tone down this sentence accordingly.

l.229: “This could explain why the EGO deletants…”, since there are alternative hypotheses.

There are various little typos and poorly-phrased sentences throughout; I recommend re-reading to edit and perhaps asking a friendly colleague or two to take a look to double-check language.

ll.299-301: I’m not sure I agree that this suggests that TORC1 interacts with FYVE domains.  Phosphosites tend to be in disordered regions of proteins (i.e., regions that are accessible to kinases), and these phosphosites are all in the disordered regions outside of the structured FYVE domains.  Since there is significant research into substrate presentation to TORC1, and those studies have not suggested a link to FYVE domains, I think this speculation could detract from the review’s focus.  (I’d also note that many of the phosphosites on these proteins are NOT near the FYVE domains, so it seems overall a somewhat tenuous argument.)

l. 354-5: The citation is missing, but in general, increased thermal stability does not suggest that glutamine impacts Pib2 folding state. Rather, this suggests that Gln directly interacts with Pib2, slightly stabilizing its folded state under high temperatures.

Author Response

I thank the reviewer for reading the manuscript in depth. I found all the comments very helpful. I believe the quality of my manuscript has now been significantly improved thanks to the reviewer’s suggestions.

Specific points:

In this timely review, Hatakeyama provides comprehensive perspective on the relationship between TORC1 and Pib2, a proposed glutamine sensor for TORC1 in yeast.  The review is straightforward to read and provides a nice historical perspective to contextualize this burgeoning field, and I think this is a timely contribution.  I have only small comments throughout that are intended to improve on the manuscript (no major revisions recommended).

Response: Thank you for the encouraging comment.

I recommend providing the full name for Pib2 in the abstract (PI3P-Binding Protein 2) for those of us less familiar with the yeast literature.

Response: This is indeed an important point. The abstract has been amended accordingly (l. 12).

Throughout, studies are described using the senior author’s name, which is certainly helpful for understanding the historical context of experiments; as a matter of principle, I think it is worthwhile to also mention the first author’s name.  So, for example, l. 113: “De Virgilio’s group” should be “Dubouloz et al. in De Virgilio’s group”; l. 163: “Maeda’s group” should be “Tanagawa and Maeda”, etc.

Response: I agree with the reviewer. I amended accordingly (now ll. 111, 129, 168, 208, 216, 238, 268, 291, 301, 412, 429, 460).

Introduction section: Given the emphasis on evolutionary comparison, I think it is worth mentioning here that Pib2 is conserved in many fungi but not well-conserved outside fungi.  I also think it is worth noting that, in plants, TOR is sensitive to amino acids (see, e.g., Schaufelberger et al., J Experimental Botany, 2019; Cao et al., eLife, 2019; O’Leary et al., Plant Cell, 2020; and Liu et al., Dev Cell, 2021) and has conserved roles in translation and ribosome biogenesis downstream from TOR (see, e.g., Xiong et al., Nature, 2013, and Scarpin et al., eLife, 2020), but that plants do not encode Gtr1/Gtr2 orthologues (see, e.g., Chantranupong et al., Cell, 2015; and Brunkard, Dev Cell, 2020).  I think this would help to strengthen the argument that there is diversity in TOR nutrient-sensing mechanisms across eukaryotes, and that the singular emphasis on Rag/Gtr GTPases is perhaps too narrow (even in humans).

Response: Thank you for this interesting and insightful comment. I lightly touched plants in the original manuscript, but now have significantly strengthened that part (now ll. 500-505).

The review often refers readers to other sections (e.g., ll. 137-138; but this happens at least ten times).  While I understand the impulse, this can make reading the review choppy: a reader should be able to read from top to bottom.  If it is important to foreshadow an in-depth discussion below, then this is OK, but in general, I don’t think it is necessary to always refer to reader to future paragraphs.  I would delete ll. 137-138, for example.

Response: Thank you for this, which is indeed important from the reader’s point of view. I deleted the majority of ‘see Section XXX’ (now ll. 125, 139, 167, 228, 363, 439). I left a few essential ones and retrospective ones.

References are occasionally missing; see, for example, l. 176, which I think should refer to citation #33.

Response: Corrected (now l.228). I also went through the manuscript again and made sure of appropriate referencing.

Repeatedly, the review refers to a citation and then also to unpublished results from the author’s group.  I recommend against this, because it potentially undercuts the published work.  Instead, this could be phrased as, “which our lab has replicated/reproduced”, if the goal is to indicate that you strongly believe a conclusion is valid.  In general, citing unpublished work or preprints in reviews seems risky to me.

Response: I agree with this opinion. I have deleted two out of three such statements (now ll. 180, 268). I left it at l. 300, because not ‘ALL these observations’ can be found in published papers.

l.176-178: I’m not sure that Pib2 is sufficient to transmit the glutamine signal. The work in citation #33 (Ukai et al., 2018, PLoS Genetics) argues that Pib2 is necessary for glutamine sensitivity and that Pib2-TORC1 interact with glutamine, but even they note that other proteins may be involved and that Pib2 may not be the sensor per se. We know from other examples in mammals that necessary components of amino acid sensor complexes are not necessarily the amino acid-interacting proteins.  I would tone down this sentence accordingly.

Response: This argument is based on the very recent paper by Tanigawa et al (now Reference 37), which I believe is valid.

l.229: “This could explain why the EGO deletants…”, since there are alternative hypotheses.

Response: Agreed and corrected (l. 294).

There are various little typos and poorly-phrased sentences throughout; I recommend re-reading to edit and perhaps asking a friendly colleague or two to take a look to double-check language.

Response: I asked my British colleague, Katy Betchetti (now acknowledged in l. 534), to do this. The corrections are scattered throughout the manuscript.

ll.299-301: I’m not sure I agree that this suggests that TORC1 interacts with FYVE domains.  Phosphosites tend to be in disordered regions of proteins (i.e., regions that are accessible to kinases), and these phosphosites are all in the disordered regions outside of the structured FYVE domains.  Since there is significant research into substrate presentation to TORC1, and those studies have not suggested a link to FYVE domains, I think this speculation could detract from the review’s focus.  (I’d also note that many of the phosphosites on these proteins are NOT near the FYVE domains, so it seems overall a somewhat tenuous argument.)

Response: I agree, the evidence for this hypothesis is not very strong. Because this part is not essential for the main theme of this review, I have deleted the whole argument (l. 369).

  1. 354-5: The citation is missing, but in general, increased thermal stability does not suggest that glutamine impacts Pib2 folding state. Rather, this suggests that Gln directly interacts with Pib2, slightly stabilizing its folded state under high temperatures.

Response: Amended with citation (now l. 446).

Reviewer 3 Report

See attached file

Author Response

I thank the reviewer for reading the manuscript in depth and providing lots of insightful comments. I found all of them very helpful. There are a few occasions where I addressed the points in a different way- these are mainly in the sake of accessibility of the paper; I intentionally avoided discussing too much yeast-specific technical details (such as strain background), because it may detract interests from non-yeast researchers. However, I do agree that these are very important factors to consider when interpreting data. On balance, instead of discussing these aspects in the main text, I newly prepared a table that includes methodological information (in line with one of the reviewer’s comments). As a result, I strongly believe that the quality and comprehensiveness of my manuscript have been significantly improved.

Specific points:

 This review deals with Pib2, a newly emerging regulator of the highly conserved TORC1 pathway. As far as I know, it is the first review completely dedicated to this protein. The project of this review is nevertheless ambitious as contradictory data are observed in the literature and the topic is emerging. I think that this review could be very useful for the large TORC1 research community. I really appreciated that the author underlines and discusses the difficulty to identify the upstream direct regulators of TORC1 as stress or nutrient transport could be affected in some deletion mutants, making the conclusions of a direct role or an indirect role of the deleted proteins more difficult. I really appreciated that this problematic has been clearly mentioned. However, I would appreciate this kind of discussion concerning other results that have to be considered more carefully, as the topic is still emerging, and the data are not complete and/or contradictory. I think that the review could be more exhaustive by considering different models explaining the research results, sometimes contradictory. For instance, the author gives a lot of importance to the “two activation phases” model while ignoring other possible explanations/models (see detailed comments below).

Response: Thank you for the overall supportive comment. I understand that there can be other models/hypotheses around the Pib2-EGO issue. I was very careful not to mislead readers as if my interpretation is the only, exclusive one (i.e., I clearly labeled in Fig 2 as ‘the authors’ speculations’). In addition, I now explicitly wrote that I ‘do not exclude other possibilities’ (l. 170). Collectively, I have taken extensive care not to mislead readers, while keeping my original insight/opinion (which I believe is important to have, as this and other reviewers appreciate).

Major comments:

- This review indicates the role of Pib2 in the TORC1 regulation after amino acid addition, but TORC1 is regulated by other signals, such as stress or nitrogen (aminoacids, ammonium) and carbon sources. It could be interesting to mention all the signals and the role of Pib2 (and eventually Gtr) in response to other signals (not only amino acids).

Response: Thank you for pointing this out. I have now strengthened these aspects (ll. 391-397).

- The two phases of TORC1 activation (l.184-263): This part of the review extensively describes a model suggesting 2 phases of TORC1 activation, and the author makes a lot of speculations about this model. Around 80 lines are associated to this model. I completely agree that it is important to mention this model and the speculations/hypotheses of the author are interesting. However, the author has to be more careful with these data as this model essentially comes from one unique publication (Stracka et al., 2014), and the results are based on one TORC1 target, Sch9 (particularly a truncated form of Sch9). If the model is right, we expect to have more phosphorylation of Sch9 in the presence of glutamine at steady-state. However, at time 0, in glutamine medium, the phosphorylation of Sch9 is minor (Stracka et al., 2014, Fig.1C). Furthermore, the total quantity of Sch9 C-ter read-out decreases after glutamine addition, notably after 30 min (also observed with other TORC1 read-outs). So the decrease of Sch9 phosphorylation could also be linked to the fact that the non-phosphorylated form is more sensitive to degradation than the phosphorylated form. I don’t say that the model is wrong, however, I want to underline that this model is based on few results that can also be explained in another way. I encourage the author to keep this part, for sure very interesting. But the part referring to this model and its interpretation could be shorted and improved by discussing these results with the results obtained by others, for instance concerning the phosphorylation of other TORC1 read-outs (Npr1, Rps6) that do not seem to follow these two activation phases. A unique paragraph explained then the role of Pib2 in these two phases. I did not expect to read a paragraph apart as it is only based on one model. I would prefer a paragraph describing in more details the contribution of Gtr and Pib2 (parallel or same pathways) in function of the different signals, lab strains and read-outs (see comment just below), for instance, a paragraph resuming the opposite results obtained by the different groups and then the proposed role of Gtr/Pib2 in the different models.

Response: In fact, I thought a lot about how much technical details I should discuss, because it may discourage non-yeast people to continue reading (many of them will not know what BY4741 is). That is why I omitted them in the initial manuscript. However, on the other hand, yeast people should certainly be interested. My solution to satisfy both cohorts was, instead of discussing within the main text, including the methodological information in the newly prepared Table 1 (as you suggested in the next comment). I think this table has made a big difference in the comprehensiveness and therefore enhanced the value of this article.

- Some groups and results suggest that Pib2 and Gtr act in the same pathway, while others suggest that they act in parallel independent pathways (Kim & Cunningham 2015; Varlakhanova et al., 2017; Michel et al, 2017; Tanigawa & Maeda, 2017; Ukai et al, 2018; Brito et al, 2019). I would have appreciated a more developed discussion of these data and particularly the description of the major results sustaining these two hypotheses, or at least the context of the experiments (signal: aminoacid?, which readout?, at short-term or steady-state?). A part of discussion is found at the end of the review, but it is one unique interpretation (“two activation phases” model) and not a clear presentation of all the available data. It could be useful in order to understand this part to mention the data sustaining the 2 models. It could also be interesting to mention that some results showed that Pib2 and Gtr1 can act on TORC1 in a different way (for instance Npr1 phosphorylation and abundance are not the same in gtr and pib2 mutants) (not only by referring to the “two activation phases” model). In this context, the reference to Hughes Hallet 2015 seems also important as they proposed that TORC1 could act as an information-processing hub, activating different genes in different conditions, so depending on the upstream signal, the signal transmitted by TORC1 could be different… - In the same vein: l.149-151: “Second, lacking either Pib2 or the EGO complex alone causes the entire loss of the acute TORC1 activation by amino acids [35][36], rather than a partial defect that the parallel model would predict.” Some data (Brito et al. 2019) showed contradictory results: deletion of Pib2 or Gtr alone causes partial defect in TORC1 activation. It could be interesting to mention. These discrepancies in results could also be explained by the use of auxotrophic strains in the first case (the 2 Pib2 and Gtr required for TORC1 activation) because, as the absence of auxotrophy (in the conditions used by Varlakhanova et al.) is a condition in which Gcn2 was proposed to inhibit TORC1 (Yuan et al., 2017). We cannot exclude that, in that case, Gcn2 inhibits TORC1 and so the simultaneous action of Gtr and Pib2 is required to reactivate TORC1. As previously mentioned, I think it is important in a review to consider all the models proposed in the literature, and the different results could also be explained by differences in the strains used by the different groups (auxotrophic or not, different background), the used read-out (Sch9, Rps6, Npr1, Gln3,…) and the signal (amino acid, N starvation, ammonium, glucose starvation, stress…). Mentioning the conditions (strain, signal, read-out) used by the different groups and their analysis in link with the opposite results obtained by the different groups will add value to this review. One figure is dedicated to the “2 activation phases” model. In the place of this figure or in addition to this figure, I would appreciate to read a table regrouping the different results sustaining the hypotheses of parallel or same Gtr/Pib2 pathway(s), and comparing the techniques used, the signal (aminoacid?, ammonium?), the genetic background (with or without auxotrophy), the read-out,… (see comment just below).

Response: Thank you for interesting ideas and hypotheses, although I cannot document all the different hypotheses within one article. I believe the addition of Table 1 addresses your points; the technical information included there will help readers develop their own hypotheses. Thank you again for your excellent suggestion, adding Table 1 has made such a big difference.

- l.209-212: The author mentions that the proton could be the signal activating TORC1. However, the author doesn’t mention the results that do not fit with this model. For instance, pH drop is often correlated to a defect of yeast growth, what does not fit with activation of TORC1 (Orij et al 2012). Acidification inhibits mTORC1 in mammalian cells (Balgi et al., 2011; Fonseca et al., 2012) and Brito et al., 2019 have shown that a pH drop is not a prerequisite for TORC1 activation upon nitrogen source addition.

Response: I have included this viewpoint (ll. 236-238).

- l.275-301: This paragraph is centered on the FYVE domain of Pib2. However, importantly, the role of this domain in the activation of TORC1 is not mentioned. Michel et al., 2017 have shown that FYVE is not absolutely required to activate TORC1.

Response: It is indeed important to mention. Added (ll. 307-309).

- Figure1 is a little bit simplistic. For instance, it mentions the activation of TORC1 but not the inhibition of the complex by Gtr and Pib2. Moreover, TORC1 is not only regulated by aminoacids but also by other signals. It is weird that, in the case of mammalian cells, another signal is mentioned (growth factor) but no other signals (N and C sources, stress) are indicated in this figure for the yeast part.

Response: My intention was to make this article focused on Pib2. I did not try to write a comprehensive review on TORC1 (many such reviews already exist). I focused on amino acids, because the mechanism by which Pib2 senses them has been extensively studied. I decided not to have C sources and stress in Fig. 1 and drawing many possible allows (which goes through or skips Pib2 and/or EGOC) and potting many ‘?’ signs, which may annoy readers. However, I found helpful your suggestion to add inhibitory arrows; this is indeed a very important aspect and worth highlighting in this Figure. I have added it now.

Minor comments:

l.9: “complex” in place of “kinase” because TORC1 is a complex comprising a kinase and not only a kinase

Response: Agreed and amended (l. 9).

l.24-25: maybe interesting to add here that rapamycin is an inhibitor of TORC1

Response: Agreed and amended (now ll. 26-27).

l.25-27: it could be interesting to mention the 2 kinases Tor1 and Tor2, as well as one sentence about the TORC2 complex.

Response: TORC2 is now mentioned (ll. 34-36). I decided not to mention Tor1/Tor2, as it is yeast-specific.

l.40: as major regulatorS

Response: Thank you for this (now l. 44).

l.42: problem in this part of sentence: “identifications of TOR itself and TORC1” because TOR is part of TORC1, maybe change with “identifications of TORC1 components” or “identifications of TOR itself and the other components of TORC1”

Response: Agreed and amended (now l. 46).

l.49: please add “called” before “the EGO complex in yeast” to make the sentence more easily readable.

Response: Agreed and amended (now l. 54).

l.53-54: “The immediate upstream regulators of Rag GTPases, e.g., GTPase-activating proteins are generally well conserved” It could be interesting to mention the name of these activating proteins if the author wants to mention their existence.

Response: Thank you, this is an excellent idea. Done it (ll. 59-61).

The review mentions in the introduction part the localization of mTORC1 in mammalian cells (l.63). However, the author does not mention the localization in yeast cells and the oligomerization and the formation of TOROID. As this structure has been involved in the inactivation of TORC1, it could be interesting to mention it. Could we imagine a role of Pib2 in these structures? Link with TORC1-body?

Response: It is touched in Section 5-3 (now ll. 357-361). I kept the argument minimum, as this is quite an important yet controversial area, which may warrant another review paper by its own.

l.68-69: I do not completely agree with “How the yeast EGO complex regulate TORC1 is unclear” (add “s” to “regulate”) because, as the author described previously, we know that the GTP-Gtr1 and GDP-Gtr2 are required to activate TORC1… This part can be deleted or reformulated. l.69: “as yeast Rheb does not seem to activate TORC1”: this second part of the sentence has no link of consequence with the previous one. “And” in place of “as”?

Response: I resentenced this part to clarify better (now ll. 77-79)

l.86 The author describes here the fact that Psr and Whi2 also regulate ammonium transport and so could influence TORC1, not directly, but indirectly by regulating ammonium transport, so the quantity of transported nitrogen source. As mentioned previously, I find very interesting to mention this point of view. However, in line with some of my comments in the introduction part, I think it could be useful to precise that ammonium, and not only amino acids, can stimulate TORC1 (maybe in the introduction paragraph). This makes sense in this review as the role of Pib2 in TORC1 activation in response to ammonium has also been studied (Brito et al. 2019). l.89: in the same vein, the author discusses the stress as a signal here. However, the author doesn’t mention the actors required for the regulation of TORC1 in response to a stress. Is Pib2 involved?

Response: Thank you for the supportive comment on the argument on nutrient transport. I also appreciate the importance of ammonium and stress, although the mechanistic aspects have not been understood as clearly as the case of amino acids. Nevertheless, I have significantly expanded the argument on these non-amino acid signals (ll. 462-471).

l.84-91: I would appreciate one or 2 sentences about the described role of Snf1 and Gcn2 in the control of TORC1. Maybe the author suspects that these proteins can affect TORC1 indirectly. However, it could be useful to at least cite these two potential upstream-TORC1 regulators in the introduction.

Response: Agreed, now mentioned (ll. 94-95).

l.95: it could be interesting to cite all the studies referring to the regulation of TORC1 by Pib2 at the end of this sentence “This 2015 work, together with the following independent studies by multiple groups including ours, has established Pib2 as a direct regulator of TORC1.”

Response: Nice idea indeed. Done (l. 113).

  1. 95-96: “The extent of Pib2’s contribution to the TORC1 activity is comparable to that of Rag GTPases.” I would delete this sentence because it is difficult to measure if their contribution is really comparable (it maybe probably depends on the signal), and this sentence is not essential in the review.

Response: Agreed, it has been deleted (l. 113).

l.127:” 2015 marked the birth year of Pib2 research.” The research on Pib2 has started before, as the author mentioned previously. The author could be more precise by mentioning research with link with TORC1, for instance: “the role of Pib2 in TORC1 regulation started to be investigated in 2015”

Response: Amended (l. 156).

l.131-132: “uncharacterized” in place of “untouched”

Response: Amended (l. 161).

l.136: between Pib2 and TORC1

Response: Thank you, amended (l. 165).

l.139 and at other places: “2015 paper” is not precise and a little bit confusing. It could be better to refer to the first or last author of the publication.

Response: Agreed, amended (ll. 111, 168, 208, 301).

l.141: I would replace “inactivation” by “absence of activation”; it’s not the same…

Response: Sentence changed (l. 170).

l.160: as previously mentioned, the role of Pib2 has also be studied in response to ammonium. It could be interesting to mention…

Response: Added (l. 212).

l.162: As far as I know, the function of Pib2 as glutamine sensor has not been proved. Results suggest a role in the sensing of glutamine (indirectly) but, as far as I know, there is no evidence that Pib2 is able to directly detect glutamine. If I’m not wrong, the glutamine sensor in mammalian cells is a transporter of glutamine. Is there any indication of a domain able to detect glutamine in the sequence of Pib2? Could we exclude that, when vacuoles are purified in the experiments described by the author, the vacuolar glutamine transporter is functional and directly responsible for glutamine sensing? Maybe next influencing the activity of the V-ATPase… Moreover, as the paper cited in this paragraph (as #) is not yet accessible, it is difficult to review this part, and the necessity to use a hyperactivating mutation in Pib2 to observe TORC1 activation could indicate an artefact… The fact that Pib2 has also an inhibiting region makes maybe this observation more complicated? Is it known if this mutation inactivates the inhibition region of Pib2?

Response: A brand-new paper (Reference 37) is now available, which addresses this point.

l.226-229: “An important finding in Hall’s study was that the EGO complex is indispensable only for Phase-1; yeast lacking the EGO complex, while showing the complete lack of the acute and transient activation, eventually activated TORC1, almost normally, 15 minutes after the amino acid stimulation.” This result could also be explained by a delay in the phosphorylation in the gtr deletant.

Response: gtr1D shows only one wave, not two. I trust their results because I was able to reproduce them by my hands (even in BY4741 background, which differs from theirs).

l.266: the “resolution” word is maybe not adapted here.

Response: Amended (l. 340).

l.319: the author mentions the conservation of the Pib2 domain. It would be interesting to know first if Pib2 is conserved or not.

Response: I totally agree, I discussed this in Section 7.

l.361-363: “As mentioned in Section 2, Npr1, a TORC1 downstream kinase, promotes Pib2 phosphorylation [31]. This suggests the existence of a feedback regulatory loop (TORC1-Npr1-Pib2-TORC1), which was confirmed by a following study [54]. What this phosphorylation does on Pib2, mechanistically, is unclear.” One fact to be added is that TORC1 inhibition by Npr1 is dependent on Pib2, so suggesting that Npr1 phosphorylation could influence the action of the inhibitory domain.

Response: Thank you, this is indeed very interesting hypothesis. Now mentioned (ll. 475-477).

l.392: “4) Does Pib2 transmit any signal other than amino acids?” not completely unknown as it was shown that ammonium is another signal transmitted by Pib2. Again, it could be interesting to mention before.

Response: Agreed, I added ‘nitrogen sources’ in a more general way (l. 522).

Round 2

Reviewer 3 Report

see attached file

Author Response

I thank the reviewer for providing additional comments and clarifications. These were really helpful in further improving the manuscript.

Specific points:

 I thank the author for the modifications he added in the review. I really appreciate the efforts made to make the review more exhaustive. I have just some minor points essentially coming from the introduced modifications:

- The author mentions that he agrees that technical details can be at the origin of the discrepancies between the different studies (cooperative or parallel actions of Pib2 and Gtr). However, he wrote that some technical details could discourage non-yeast people to continue reading. I agree that all technical details, such as the name of the yeast background, are not useful. However, I think it could be useful for all to mention for instance that the absence of auxotrophy (in the conditions used by Varlakhanova et al. for instance) is a condition in which Gcn2 was proposed to inhibit TORC1 (Yuan et al., 2017) so this particular context could add a supplemental upstream regulator and could explain why the simultaneous action of Gtr and Pib2 is required to reactivate TORC1. This constitutes an alternative hypothesis to the one proposed by the author and would probably also interest non-yeast scientists as Gcn2 is a conserved protein.

Response: I agree, this specific point is now mentioned (ll.166-169).

- I thank the author to have added Table 1 that I found very useful. To be more exhaustive, the author could add two other references in this table:

¢ Tanigawa & Maeda, 2017: this publication is interesting in this context as the requirement of Gtr and Pib2 for TORC1 activation seems to be different depending on the signal (glutamine or arginine)

¢ Brito et al., 2020: in this publication, another TORC1 target (Npr1 in addition to Rps6) was used and the genetic background is different (no amino acid added to complete auxotrophy).

Response: Two papers were added to Table 1. The glutamine-arginine issue is now mentioned in the main text (ll.187-189).

- I think the author did not catch one of my previous points concerning the activation of TORC1 in gtr mutant. o My past comment: l.226-229: “An important finding in Hall’s study was that the EGO complex is indispensable only for Phase-1; yeast lacking the EGO complex, while showing the complete lack of the acute and transient activation, eventually activated TORC1, almost normally, 15 minutes after the amino acid stimulation.” This result could also be explained by a delay in the phosphorylation in the gtr deletant.

o Response of the author: gtr1D shows only one wave, not two. I trust their results because I was able to reproduce them by my hands (even in BY4741 background, which differs from theirs).

My new comment: For sure, I didn’t write that I didn’t trust the data. What we can observe in the western blot is a phosphorylation that appears later in gtr1 compared to Wt cells. I wanted to mention that what the author considers as only one wave (corresponding to Phase 2) in gtr1D, it could just correspond to a delay in the activation of TORC1 (delayed Sch9 phosphorylation) in the gtr mutant because for instance the signal is not so efficiently transmitted. This is an alternative explanation to the 2-phases model proposed by the author. Moreover, this 2-phases activation model does not seem to be observed with other TORC1 targets, such as Npr1. This result was not mentioned by the author and suggests that an alternative hypothesis would be worth mentioning.

Response: These are important viewpoints indeed. Now added (ll.169-171 and ll.290-295).